# A manganese(I) complex with a 190 ns metal-to-ligand charge transfer lifetime

Sandra Kronenberger ⬡, Robert Naumann ⬡, Christoph Förster ⬡, Nathan R. East, Jan Klett ⬡ & Katja Heinze ⬡ ✉

Application of photoactive transition metal complexes in lighting, imaging, sensing, and photocatalysis is usually based on the triplet metal-to-ligand charge transfer ($^3$MLCT) excited state of precious metal complexes with 4/5d$^6$ valence electron configurations. These photocatalysts exhibit excited state lifetimes exceeding hundreds of nanoseconds. Simple 3d$^6$ transition metal complexes containing abundant metals exhibit lifetimes below 1–2 nanoseconds, and they require multistep ligand syntheses mitigating large-scale implementation. We report that a commercially available bis(imidazolium) pyridine pro-ligand [$H_2$pbmi]$^{2+}$ and a manganese(II) salt yield the tetracarbene manganese(I) complex [Mn(pbmi)$_2$]$^+$. This complex phosphoresces at room temperature in fluid solution from a $^3$MLCT state with a lifetime of 190 ns. In combination with the reversible [Mn(pbmi)$_2$]$^{2+/+}$ process, this translates to an excited state capable of reducing benzophenone. Combination of manganese(I) with rigid carbene/pyridine ligands expands key strategies for photoactive 3d$^6$ metal complexes of earth-abundant metals with $^3$MLCT lifetimes rivalling those of precious metals and providing a conceptual starting point for a sustainable photochemistry.

Precious metal complex photosensitizers, e.g. based on ruthenium(II) or iridium(III) with 4d$^6$ and 5d$^6$ electron configuration, respectively, possess strong visible light absorbance and long triplet metal-to-ligand charge transfer ($^3$MLCT) state lifetimes with hundreds of nanoseconds[1,2]. Hence, these benchmark complexes play key roles in photochemical synthesis, energy-efficient lighting, and photodynamic therapy[3–9]. However, the high cost and insufficient availability of suitable light-harvesting materials based on precious metals mitigates a widespread implementation, which would be required to power reactions with light such as large-scale photochemical organic synthesis and solar fuel synthesis. Hence, novel concepts towards a sustainable photochemistry with abundant elements are strongly sought after[10–15]. Abundant non-noble metal ions with 3d$^6$ electron configuration, such as iron(II), typically suffer from ultrafast relaxation of the $^3$MLCT states via low-energy metal-centered (MC) excited states to the ground state (GS)[16,17] preventing efficient photochemical and photovoltaic applications. Clever ligand design strategies using chelating isonitrile, carbene, and cyclometalating ligands had increased the

$^3$MLCT lifetime of 3d$^6$ metal complexes (Cr$^0$, Mn$^I$, Fe$^{II}$) from the sub-picosecond regime up to 1–2 ns in recent years[18–29]. Decoration of polyisonitrile chromium(0) complexes with six mesityl or pyrenyl substituents even raised the lifetime to 47 ns[19]. Yet, the photophysical key values of these synthetically sophisticated and large complexes are still an order of magnitude lower than those of their precious metal counterparts. In addition, their syntheses require toxic and expensive low-valent carbonyl starting materials, e.g. MnBr(CO)$_5$, toxic or expensive reducing reagents, e.g. sodium amalgam or cobaltocene and furthermore multistep syntheses of the chelating ligands, e.g. substituted polyisonitriles[18–21].

N-heterocyclic carbenes in combination with electron accepting pyridines had increased the $^3$MLCT lifetime of iron(II) complexes from the femtosecond to the picosecond range. The tetracarbene complex [Fe(pbmi)$_2$]$^{2+}$ had been a key inspiration (pbmi = (pyridine-2,6-diyl) bis(3-methylimidazol-2-ylidene))[22,28], and lifetimes with several dozens of picoseconds have been achieved with iron(II) based on carbene/pyridine ligand motifs[23–27]. Isoelectronic manganese(I) complexes

Department of Chemistry, Johannes Gutenberg University Mainz, Mainz, Germany. ✉e-mail: katja.heinze@uni-mainz.de

coordinated to oligodentate ligands with only carbene and pyridine donors, however, are lacking. Merely, a heteroleptic thermally labile carbonyl pbmi manganese(I) complex [Mn(CO)$_3$(pbmi)]$^+$, prepared from MnBr(CO)$_5$ is known[30]. Yet, this complex with *trans* coordinated carbonyl ligands is very labile and its photophysical properties were not reported[30]. In fact, carbonyl manganese(I) complexes are rather used as CO-releasing molecules[31]. Hence, carbonyl ligands, in particular when coordinated in *trans* position[32,33], appear unsuitable for the preparation of carbene manganese(I) complexes with long $^3$MLCT lifetimes due to their lability. Additionally, the often employed MnBr(CO)$_5$ as starting material is unsuited for the preparation of manganese(I) with merely carbene and pyridine donors as replacement of all carbonyl ligands by carbene and pyridine fails. A different synthetic access towards carbene/pyridine complexes of manganese(I), avoiding carbonyl ligands, is clearly warranted.

Here, we identify the tridentate dicarbene pyridine pincer ligand pbmi as ideal framework for 3d$^6$ manganese(I) ions providing stability via the chelate effect and a high metal-ligand bond covalence, reducing large-amplitude distortions in the excited states and pushing the detrimental MC states to higher energy. The complex [Mn(pbmi)$_2$]$^+$ is straightforwardly prepared from easily available non-toxic, carbonyl-ligand-free starting materials, possesses a luminescent $^3$MLCT state with a record excited state lifetime of 190 ns for 3d$^6$ metal complexes, high photostability, reversible redox chemistry, and an excited state redox potential suitable for reducing organic substrates.

## Results and discussion
### Synthesis and ground state properties

The readily-accessible[34,35] and even commercially available methyl-substituted bis(imidazolium) pro-ligand [H$_2$pbmi]Br$_2$ (CAS number

263874-05-1) was deprotonated to the biscarbene with sodium bis(trimethylsilyl)amide Na[N(SiMe$_3$)$_2$] and coordinated to manganese(II) triflate Mn[OTf]$_2$ as convenient manganese(II) source in a straightforward one-pot synthesis. Excess of the carbene ligand pbmi serves to reduce manganese(II) to the desired manganese(I) complex [Mn(pbmi)$_2$]$^+$ in situ so that no further reducing agent such as sodium amalgam[18,19] is required (Fig. 1a). The facile reduction of Mn$^{II}$ by the carbene[36] pbmi was corroborated spectroscopically by treating the independently prepared [Mn(pbmi)$_2$]$^{2+}$ complex with pbmi (Supplementary Fig. 1).

$^1$H and $^{13}$C{$^1$H} NMR spectra as well as electrospray ionization (ESI$^+$) mass spectra of [Mn(pbmi)$_2$][OTf] confirm the proposed chemical structure with meridional geometric configuration and the low-spin 3d$^6$ electron configuration of the diamagnetic complex cation [Mn(pbmi)$_2$]$^+$ (Supplementary Figs. 2–4). Crystallization of [Mn(pbmi)$_2$][OTf] from THF/Et$_2$O yielded dark purple crystals suitable for X-ray diffraction (XRD) analysis (Fig. 1b). The Mn–C and Mn–N bond lengths below 2 Å substantiate the low-spin character of the manganese(I) center. Hence, [Mn(pbmi)$_2$]$^+$ is isoelectronic to the low-spin 3d$^6$ iron(II) complex [Fe(pbmi)$_2$]$^{2+}$[22]. Interestingly, the Mn–C/N bond lengths of [Mn(pbmi)$_2$][OTf] are very similar to the Fe–C/N distances in the iron(II) complex (CCDC-915795), in spite of the different formal oxidation states of the metal ions[22]. The reduced ionic character of the metal-ligand bonds in [Mn(pbmi)$_2$]$^+$ compared to the iron(II) analogue seems to be compensated by a higher Mn–C/N bond covalency.

Infrared (IR) and Raman spectra with excitation at 1064 nm of [Mn(pbmi)$_2$][OTf] show bands for vibrations of the MnC$_4$N$_2$ coordination sphere and the pyridine rings up to 1000 cm$^{-1}$ and modes involving the heterocyclic donors up to 1600 cm$^{-1}$ (Fig. 1c, Supple-

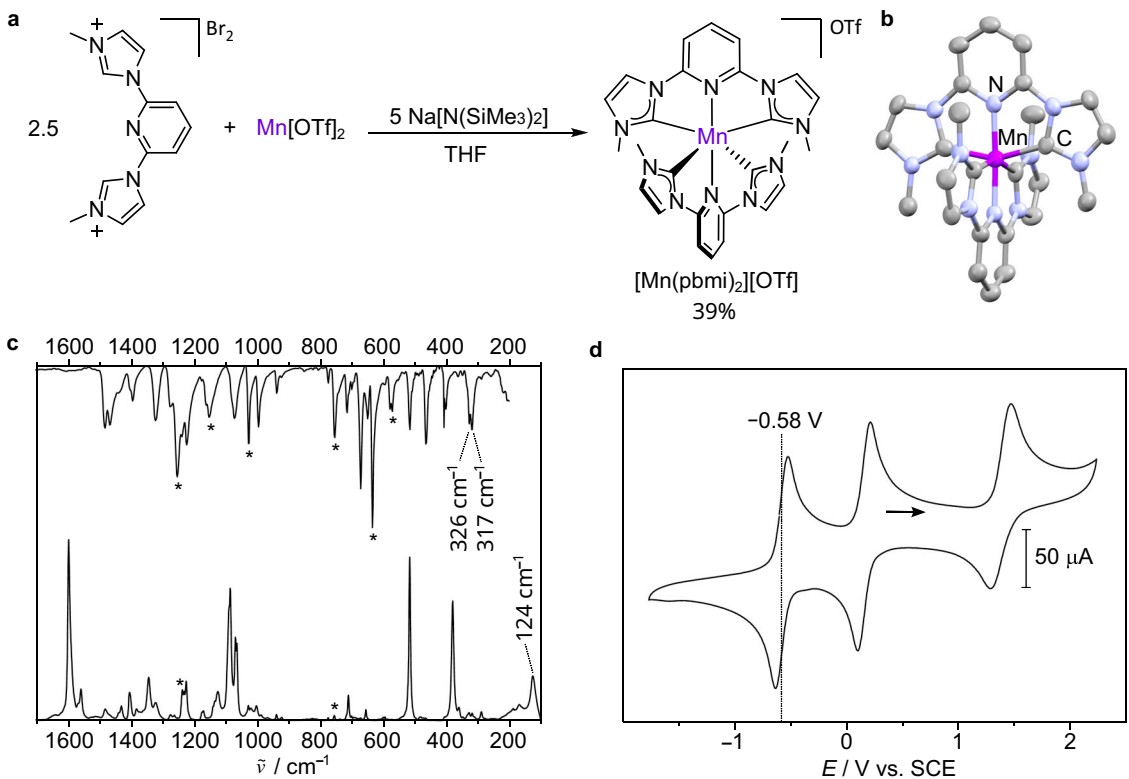

**Fig. 1 | Synthesis, structure, and ground state properties of [Mn(pbmi)$_2$][OTf].**
**a** Synthesis of [Mn(pbmi)$_2$][OTf] from Mn[OTf]$_2$, excess pro-ligand [H$_2$pbmi]Br$_2$, and Na[N(SiMe$_3$)$_2$] in THF. **b** Molecular structure of the cation of [Mn(pbmi)$_2$][OTf] determined by single-crystal X-ray diffraction (XRD), shown with thermal ellipsoids at the 50 % probability level and hydrogen atoms omitted for clarity. **c** Infrared (IR)

(top) and Raman (bottom, excitation at 1064 nm) spectra of [Mn(pbmi)$_2$][OTf] in the solid state. Asterisks denote bands of the counter ion. **d** Cyclic voltammogram of [Mn(pbmi)$_2$][OTf] (1 mM) in CH$_3$CN containing [$^n$Bu$_4$N][PF$_6$] (100 mM) as supporting electrolyte. Scan rate 100 mV s$^{-1}$. Potentials are referenced vs SCE[37].

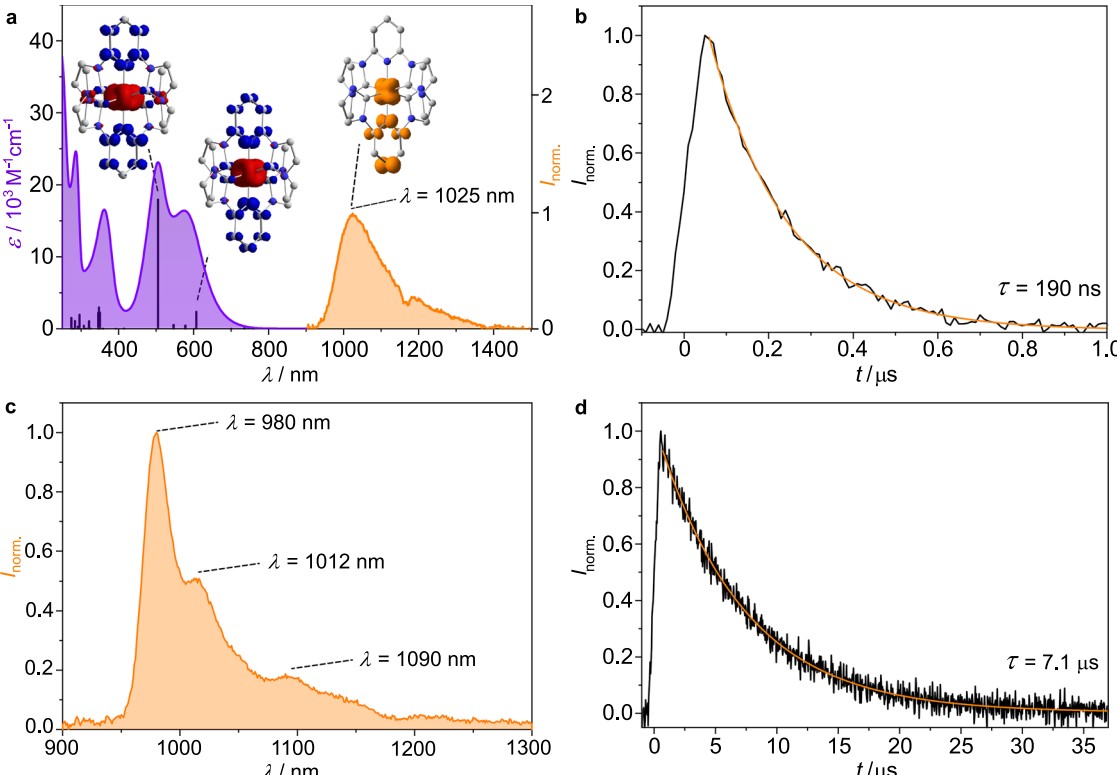

**Fig. 2 | Absorption as well as steady-state and time-resolved emission spectroscopy. a** UV-vis absorption spectrum of [Mn(pbmi)₂][OTf] in CH₃CN (left, purple), TDDFT-calculated oscillator strengths (black, shifted by 0.33 eV to lower energies; B3LYP/Def2-TZVP/CPCM(acetonitrile)/D3BJ) and luminescence spectrum of [Mn(pbmi)₂][OTf] in CH₃CN after excitation with $\lambda_{exc}$ = 450 nm at 293 K (cw laser, 1089 mW, right, orange). The "dip" at 1166 nm arises from CH₃CN overtone absorptions (Supplementary Fig. 9). Left inset: TDDFT calculated electron density difference maps of geometry-optimized [Mn(pbmi)₂]⁺ showing electron density gain (blue) and depletion (red) in the ¹MLCT(11) and ¹MLCT(5) Franck-Condon states (isosurface at 0.003 a.u., H atoms omitted for clarity). Right inset: DFT-optimized geometry and spin density surface (orange, isosurface at 0.01 a.u., H atoms omitted for clarity) of the lowest energy triplet state of [Mn(pbmi)₂]⁺ (³MLCT(1)). **b** Luminescence decay trace of [Mn(pbmi)₂][OTf] at 1025 nm in dry, deaerated CH₃CN at 293 K with excitation at 450 nm. A monoexponential fit to the data is shown in orange. **c** Luminescence spectrum of [Mn(pbmi)₂][OTf] in dry, deaerated 2-MeTHF after excitation with $\lambda_{exc}$ = 505 nm at 77 K (orange). **d** Luminescence decay trace of [Mn(pbmi)₂][OTf] at 980 nm in dry, deaerated 2-MeTHF at 77 K with excitation at $\lambda_{exc}$ = 450 nm. A monoexponential fit to the data is shown in orange.

mentary Fig. 5). In the far-IR spectral region, a strong double absorption band at 317/326 cm⁻¹ is assigned to an antisymmetric N–Mn–N stretching mode as calculated by density functional theory (DFT) methods (324 cm⁻¹; scaled by 0.98). In the Raman spectrum, the lowest frequency vibration observed at 124 cm⁻¹ is assigned to a symmetric breathing mode of the coordination polyhedron that can be described as pincer-like motion of both tridentate ligands according to DFT calculations (118 cm⁻¹) on the cation [Mn(pbmi)₂]⁺. These symmetric and antisymmetric vibrations of the MnC₄N₂ coordination polyhedron will be relevant for the excited state dynamics (see section "Photoluminescence and excited state dynamics").

The manganese(I) complex [Mn(pbmi)₂]⁺ is reversibly oxidized to [Mn(pbmi)₂]²⁺ at $E_{1/2}$(Mnᴵᴵ/ᴵ) = −0.58 V vs SCE[37] in CH₃CN/[ⁿBu₄N][PF₆] (Fig. 1d). Mnᴵᴵᴵ/ᴵᴵ and Mnᴵⱽ/ᴵᴵᴵ oxidation processes are observed at +0.16 and +1.37 V, while a reductive event lies outside the solvent potential window (ca. −2.6 V, Supplementary Fig. 6). Compared to the Mnᴵᴵ/ᴵ redox couple of polyisonitrile manganese(I) complexes at ca. +1.0 V vs SCE[20], the present Mnᴵᴵ/ᴵ process occurs at much more negative electrochemical potentials.

**Electronic spectroscopy**

The steady-state UV-vis absorption spectrum of the dark purple [Mn(pbmi)₂][OTf] salt in CH₃CN displays an intense double band spanning almost the entire visible spectral region with maxima peaking at $\lambda_{max}$ = 505 and 575 nm ($\varepsilon$ = 23,000 and 16,500 M⁻¹ cm⁻¹, respectively; Fig. 2a). Both absorption bands of [Mn(pbmi)₂]⁺ weakly shift

solvatochromatically from 505/575 nm in CH₃CN to 509/579 nm and 511/580 nm in THF and CH₂Cl₂, respectively (Supplementary Fig. 7). Excitation at 585, 633 and 785 nm resonantly enhances distinct vibrational modes at 655, 1002 and 1068 cm⁻¹ with different enhancement factors pertaining to the different excitation energies relative to the Raman absorption bands of the off-resonance Raman spectrum with excitation at 1064 nm (Supplementary Fig. 8). All vibrational bands correspond to in-plane pyridine deformation modes with the latter two being strongly coupled with in-plane carbene deformations according to DFT calculations (656, 1002, 1066 cm⁻¹). The intense absorption bands with resonance enhancement of pyridine vibrational modes are consistent with a dominant manganese-to-pyridine charge transfer character of the transitions. Time-dependent density functional theory (TDDFT) calculations on [Mn(pbmi)₂]⁺ support this assignment. Both intense absorption bands arise from spin-allowed metal-to-ligand charge transfer transitions with electron density symmetrically shifting from manganese to both pyridines (¹MLCT, Fig. 2a, Supplementary Table 1). Ligand-to-ligand charge transfer (¹LLCT) character also contributes to these excited states, but to a smaller extent (¹MLCT(11) and ¹MLCT(5) with 53.6 %/29.8 % and 53.9 %/27.6 % MLCT/LLCT character, respectively; Supplementary Table 1).

The intense bands are significantly bathochromically shifted relative to the corresponding bands of the yellow-brown iron(II) analogue ($\lambda_{max}$ = 390 and 457 nm)[22] and the yellow manganese(I) complexes with polyisonitrile ligands ($\lambda_{max}$ = 385/395 nm)[20]. Compared with the iron(II) complex[22], the higher energy d orbitals of

manganese(I) explain its lower energy MLCT states. Compared with the polyisonitrile manganese(I) complexes[20], the lower energy π* orbital of the pyridine explains the lower energy MLCT states. Beyond these intense ¹MLCT(5) and ¹MLCT(11) transitions calculated at 607 and 505 nm (shifted by 0.33 eV to lower energy), the TDDFT calculations identify four ¹MLCT transitions with very low oscillator strengths at lower energy than the intense ¹MLCT transitions (dark states ¹MLCT(1) – ¹MLCT(4) between 760 and 700 nm; Supplementary Table 1). The different resonance enhancement in the Raman spectra with 785 and 633 nm excitation (Supplementary Fig. 8) confirms the presence of weakly absorbing states with ¹MLCT character in this spectral region. These dark ¹MLCT states will be relevant for the radiative decay of the complex (see section "Photoluminescence and excited state dynamics").

## Photoluminescence and excited state dynamics

Excitation of a CH₃CN solution of [Mn(pbmi)₂][OTf] with 450 nm at 293 K results in a broad unstructured emission band peaking at 1025 nm (1.21 eV, Fig. 2a). The energy difference between lowest-energy absorption maximum (¹GS → ¹MLCT) and emission maximum (³MLCT → ¹GS) amounts to $\Delta E = 0.94$ eV suggesting a major reorganization after the excitation and/or the presence of dark states in-between. The dark states ¹MLCT(1) – ¹MLCT(4) at lower energies than the bright ¹MLCT(5) state found by TDDFT calculations (see section "Electronic spectroscopy") in part account for this large energy difference $\Delta E$.

The near-infrared photoluminescence decays monoexponentially with $\tau_{PL} = 190$ ns in CH₃CN at 293 K (Fig. 2b). This value surpasses the ³MLCT lifetimes of all previously reported 3d⁶ transition metal complexes by more than one to two orders of magnitude[16,18–29] and approaches the value of the benchmark precious metal complex [Ru(bpy)₃]²⁺ (bpy = 2,2'-bipyridine, $\tau_{PL} = 744$–877 ns at 298 K)[1,38].

In spite of the exceptionally long lifetime, the luminescence quantum yield at 293 K in solution is estimated as only $\Phi_{PL} < 10^{-5}$. Hence, the phosphorescence rate constant is very small with $k_{p,exp} = \Phi_{PL} / \tau_{PL} < 100$ s⁻¹, in the range of symmetry- and spin-forbidden spin-flip transitions[14,39]. This unexpected observation is rationalized by quantum chemical calculations. SOC-TDDFT calculations on the DFT-optimized ³MLCT state reveal only a very small singlet admixture to the three ³MLCT(1) sublevels, which is based on weak spin-orbit coupling (SOC, Supplementary Tables 2–6). Together with the small ³MLCT(1) – ¹GS energy gap, this leads to a small calculated phosphorescence rate constant $k_{p,DFT} = 35$ s⁻¹ according to the Strickler-Berg relationship[40–42] (Supplementary Table 2) in good agreement with the experimental estimation.

In a glassy 2-MeTHF matrix at 77 K, the emission band develops a vibrational fine structure with maxima at 980, 1012 and 1090 nm (Fig. 2c). Excitation spectra with observation at 980 and 1020 nm follow the absorption spectrum between 300 and 800 nm confirming that the observed emission bands arise from [Mn(pbmi)₂]⁺ (Supplementary Fig. 10–11). The energy difference between the first vibrational maxima amounts to ca. 323 cm⁻¹. This corresponds to the antisymmetric N–Mn–N vibration on the ¹GS potential according to the DFT frequency calculations (see section "Synthesis and ground state properties", 324 cm⁻¹) and the experimental IR data (317/326 cm⁻¹) (Fig. 1c, Supplementary Fig. 5). Clearly, the geometry relaxed excited ³MLCT(1) state is unsymmetrically distorted relative to the ¹GS geometry. The already long room temperature luminescence lifetime of [Mn(pbmi)₂]⁺ further increases to 7.1 μs at 77 K in the frozen matrix (Fig. 2d). Consequently, thermally activated non-radiative decays are further retarded. The combined data are fully consistent with phosphorescence from a ³MLCT state that is slightly distorted with respect to the more symmetric singlet ground state ¹GS.

DFT optimization of the lowest energy triplet state ³MLCT(1) of [Mn(pbmi)₂]⁺ shows that the spin density is distributed over the

manganese ion and a single pyridine ring suggesting metal-to-ligand character with a hole localized at the metal and an electron at a single pyridine, respectively (Fig. 2a, spin density surface). The Mn–N bond to the pyridine radical anion is longer by ca. 0.02 Å. While the initially excited ¹MLCT states (Franck-Condon states ¹MLCT(5) and ¹MLCT(11), Fig. 2a, calculated electron density difference maps) display an excess electron density delocalized over both pyridines, the relaxed ³MLCT(1) state possesses spin density localized on a single pyridine. The calculated distortion of the ³MLCT(1) state is fully consistent with the observed vibrational progression assigned to asymmetric N–Mn–N modes (see section "Synthesis and ground state properties"). The observation that the Mn–N distance to the pyridine radical anion is longer than the Mn–N distance to the neutral pyridine suggests that π-backbonding from the metal to the ligand is diminished in the former bond. π-backbonding also implies a significant covalent character to the bonds between manganese(I) and the pbmi ligands in the ¹GS. Indeed, ³MC states (as described by a single electron transfer from a $d_{xz}$, $d_{yz}$ or $d_{xy}$ orbital to a $d_{z^2}$ or $d_{x^2-y^2}$ orbital; see Supplementary Table 1 for corresponding ¹MC states, transitions 12 and 13 both with 45.9 % MC character) could not be localized by DFT calculations even with induced distortions along the manganese-ligand bonds (Mn–N up to 2.5 Å; Mn–C up to 3.0 Å). Instead, a distorted triplet charge-transfer state (³CT) with a Mn–N bond length of 2.376 Å described by a manganese(II) intermediate spin ($t_{2g}^4 e_g^1$) antiferromagnetically coupled to a ligand radical could be localized 0.59 eV above the lowest energy ³MLCT state. In addition to the high energy, the doubly excited configuration of this ³CT state prevents strong coupling with the ground state. Hence, this state will not contribute to the excited state decay. The combination of strong covalent metal-ligand bonds and the rigidity of the chelate ligands appear to prevent distorted low-energy ³MC states. Together with the low energy ³MLCT states thanks to the high energy d orbitals of manganese(I), the ³MLCT/³MC energy gap is large and excited state decay via ³MC states is of little importance.

The excited state dynamics of [Mn(pbmi)₂]⁺ on the nanosecond timescale as observed by the photoluminescence decay ³MLCT(1) → ¹GS is hence clear-cut without multiple decays as had been observed for flexible polyisonitrile manganese(I) complexes with conformational isomers being present in solution[20].

To obtain deeper insights into the ultrafast dynamics before reaching the emissive ³MLCT(1) state, fs-ns-transient absorption spectra were measured for [Mn(pbmi)₂][OTf] in CH₃CN solution at 293 K (Fig. 3, Supplementary Fig. 12). The spectral evolutions over different time scales after excitation at 505 nm are depicted in Fig. 3a, b. The dominant negative bands in the 465–650 nm region correspond roughly to the inverted steady-state absorption spectrum and are attributed to the ground-state bleach (GSB). Excited state absorptions (ESAs) initially peak on the blue and red side of the GSB at 412 and 692 nm, respectively (Fig. 3a). With a lifetime of $\tau_1 = 210$ fs these bands shift to 430 and 705 nm, respectively. By extending the time window to the pico- and nanosecond range (Supplementary Fig. 12), two additional processes, $\tau_2 = 9$ ps and $\tau_3 = 31$ ps, are observed. Time-resolved transient absorption spectra on the ns–μs timescale (Fig. 3b) show a monoexponential decay of the ³MLCT spectrum without further spectral changes fully recovering the ground state with a lifetime $\tau_4 = 190$ ns. This value corroborates the record ³MLCT photoluminescence lifetime obtained by the time-resolved emission measurements (Fig. 2b). The full ground state recovery on long time scales confirms the absence of irreversible photochemistry.

In the very early time window monitoring the ultrafast excited state dynamics, two quasi-isosbestic points at 404 and 735 nm are identified (Fig. 3a), pointing to a state-to-state transition. The amplitudes of the GSB are nearly constant during the first 500 fs, indicating that this transition does not involve repopulation of the ¹GS. In contrast, relaxation back to the ¹GS takes place on nanosecond time scales (Fig. 3b), which is reflected in identical decay profiles of the GSB and

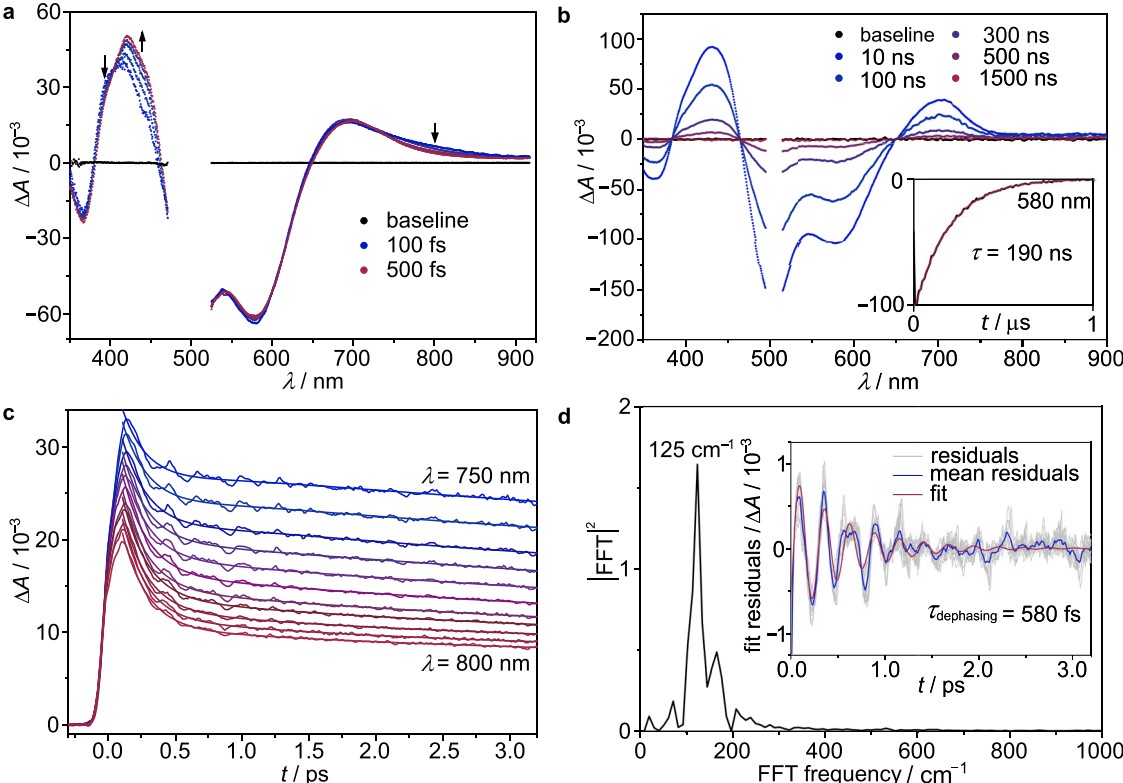

**Fig. 3 | Transient absorption spectroscopy. a** Pump-probe transient absorption spectra of [Mn(pbmi)$_2$][OTf] in dry, deaerated CH$_3$CN at 293 K after excitation with 505 nm laser pulses in the 100–500 fs range. **b** Pump-probe transient absorption spectra of [Mn(pbmi)$_2$][OTf] in dry, deaerated CH$_3$CN at 293 K after excitation with 505 nm laser pulses in the 10–1500 ns range. Inset, corresponding decay trace (black) at 580 nm superimposed with monoexponential fit (red). **c** Decay traces between 750 nm and 800 nm (5 nm steps, red to blue) and the corresponding triexponential fits. **d** Residuals of the individual traces (gray), mean residuals (blue), and the fit of the mean residuals obtained with an exponentially decaying cosine function (red). Main plot, FT spectrum obtained by fast Fourier transformation (FFT) of the wavepacket oscillations revealed by the fit residuals displayed in the inset.

ESAs as well as three isosbestic points at 385, 464, and 651 nm at zero differential absorption. Evolution associated difference spectra (Supplementary Fig. 12) obtained from the transient absorption spectra after global analysis demonstrate that the spectral signature changes significantly at early times ($\tau_1$). This indicates a change in the type of electronic state. On the other hand, the processes on the picosecond timescale are likely associated with cooling and solvent reorganization in the triplet manifold as the spectral shape is largely preserved. The ultrafast process ($\tau_1$) with a change in electronic nature can either correspond to intersystem crossing (ISC) from $^1$MLCT(n) to $^3$MLCT(m) states or to internal conversion (IC) between $^3$MLCT(m) states after an ultrafast ISC process, which would not have been resolved with our instrument.

A closer look at the fit residuals of the transient absorption data for [Mn(pbmi)$_2$][OTf] after excitation at 505 nm revealed coherent oscillations at short delay time superimposed on the initial multi-exponential kinetics (Fig. 3c, Supplementary Figs. 13–15). The oscillations are most pronounced in the spectral range of 750–800 nm and decay with a dephasing lifetime of $\tau_{dephasing} = 580$ fs (Fig. 3d). A fast Fourier transformation (FFT) analysis of the coherent oscillations displayed in Fig. 3d yields a frequency at 125 cm$^{-1}$ (Fig. 3d). Comparable coherent oscillations had been observed for copper(I) and iron(II) complexes (125–215 cm$^{-1}$)[43–47], but never for manganese(I) complexes. These coherent oscillatory features for copper(I) and iron(II) complexes were assigned to symmetric breathing modes of the coordination polyhedron in the respective triplet excited states. Based on the experimental ground state Raman spectrum with a low-frequency Raman band at 124 cm$^{-1}$ (see section "Synthesis and ground state properties", Fig. 1c, Supplementary Fig. 5), we assign this mode to symmetric pincer-type breathing of the MnC$_4$N$_2$ coordination polyhedron of the excited state in agreement with the symmetric modes previously assigned to copper(I) and iron(II) complexes[43–47]. Consequently, these oscillations observed for [Mn(pbmi)$_2$][OTf] originate from a manganese-ligand stretching vibrational wavepacket on the excited state potential.

Overall, the combined data suggest a very fast ISC process, a symmetric expansion of the coordination sphere, followed by cooling and localization within the triplet manifold to arrive at a long-lived, slightly distorted $^3$MLCT(1) state with a longer Mn–N bond to the pyridine radical anion of pbmi$^{\cdot-}$ and four elongated Mn–C bonds involving both chelate ligands. Contrasting the excited state evolution via distorted low-energy $^3$MC states of the analogous iron(II) complex [Fe(pbmi)$_2$]$^{2+}$ [20,48], $^3$MC states appear to be not involved in the excited state decay of the long-lived $^3$MLCT(1) state. Indeed, at the ground state geometry, the unoccupied σ-antibonding Mn 3d$_{z2}$ or 3d$_{x2-y2}$ orbitals are 2.6 and 3.0 eV, respectively, above the LUMO, which is essentially composed of pyridine π* orbitals (Supplementary Table 7). The large ligand field splitting due to a strong covalency of the manganese(I)-ligand bonds contrasts the isoelectronic [Fe$^{II}$(pbmi)$_2$]$^{2+}$ complex, which possesses a distorted $^3$MC state at low energy[48]. A large-amplitude N–Mn–N elongation that would stabilize $^{1/3}$MC states with a single 3d$_{z2}$ occupation is restricted in the manganese(I) analogue due to the rigid tridentate chelate with strong Mn–C bonds. Consequently, $^3$MC states do not appear to be relevant for the non-radiative excited state decay of [Mn(pbmi)$_2$]$^+$. The negligible accessibility of $^3$MC states of [Mn(pbmi)$_2$]$^+$ also accounts for its high photostability in CH$_3$CN with a photodegradation quantum yield $\Phi_{deg} = 0.0002$ %, which even surpasses that of [Ru(bpy)$_3$]$^{2+}$ ($\Phi_{deg} = 0.022$ %)[32] by two orders of magnitude (Supplementary Fig. 16). The experimentally observed long $^3$MLCT state lifetime and the

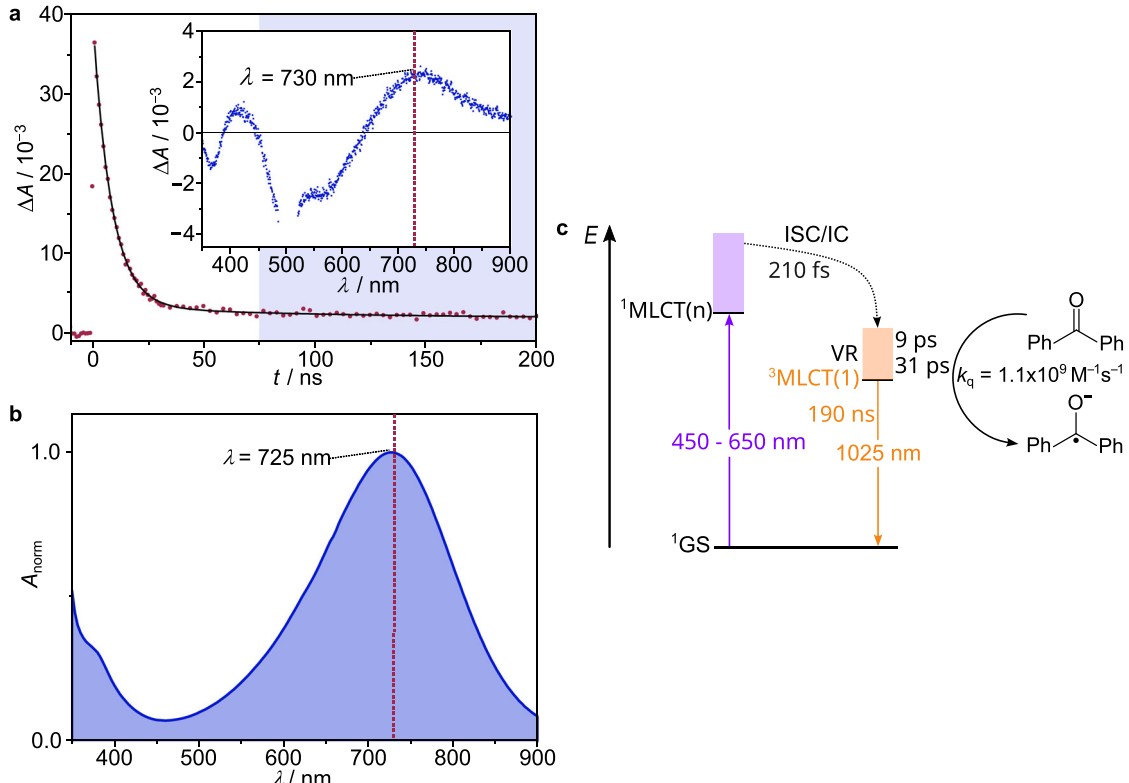

**Fig. 4 | Photoinduced electron transfer. a** Inset: Transient absorption spectrum of [Mn(pbmi)$_2$][OTf] (280 μM) in dry, deaerated CH$_3$CN in the presence of benzophenone (100 mM) averaged over 75–280 ns (blue) after the excitation pulse ($\lambda_{exc}$ = 505 nm). Main: Corresponding decay trace (red) at 730 nm superimposed with monoexponential fit (black). **b** UV-vis-NIR absorption spectrum of BP$^{•-}$ generated by electrochemical reduction of BP (6.2 mM) in CH$_3$CN containing [$^n$Bu$_4$N]

[PF$_6$] (100 mM) as supporting electrolyte. **c**, Energy-level scheme for the manganese(I) complex, colored arrows represent light absorption and emission. The curved dotted black arrow comprises intersystem crossing (ISC), internal conversion(s) (IC), and vibrational relaxation (VR). The curved black arrow denotes electron transfer to benzophenone BP.

photostability allow the exploitation of excited [Mn(pbmi)$_2$]$^+$ in bimolecular quenching.

**Bimolecular photoinduced electron transfer**

Beyond the long excited state lifetime and photostability, the excited state redox potential $^*E_{1/2}$(Mn$^{II/I}$) defines the substrate scope for photoinduced electron transfer. With the $^1$GS redox potential of $E_{1/2}$(Mn$^{II/I}$) = −0.58 V vs SCE, the energy gap between the lowest vibrational levels of the $^3$MLCT(1) and $^1$GS states $E_{00}$ = 1.30 eV as determined from the emission spectrum at 77 K and neglecting the Coulomb term, the present manganese(I) complex possesses an excited state redox potential of $^*E_{1/2}$(Mn$^{II/I}$) = $E_{1/2}$(Mn$^{II/I}$) − $E_{00}$ = −1.88 V vs SCE. This negative reduction potential of the $^3$MLCT state should be sufficient to reduce benzophenone (BP) to its radical anion BP$^{•-}$ ($E_{1/2}$(BP$^{0/•-}$) = −1.83 V vs SCE)[49,50]. Excitation of [Mn(pbmi)$_2$][OTf] in the presence of 100 mM BP in CH$_3$CN dramatically reduces the excited state lifetime from $\tau_4$ = 190 ns to $\tau_{BP}$ = 8.5 ns (Fig. 4a, Supplementary Fig. 17), giving an estimated Stern-Volmer quenching constant $K_{SV}$ = 210 M$^{-1}$ and a quenching rate constant $k_q$ = 1.1 × 10$^9$ M$^{-1}$ s$^{-1}$. In order to confirm the photoreaction product, a nanosecond transient absorption spectrum was recorded over 75–280 ns after the excitation. This spectrum displays a characteristic band maximum at ca. 730 nm (Fig. 4a). Apart from a small shift caused by the GSB, this band matches the absorption maximum of an authentic BP$^{•-}$ sample ($\lambda$ = 725 nm) prepared by electrochemical reduction of BP (Fig. 4b). Yet, the intensity of the BP$^{•-}$ absorption suggests that cage escape is very inefficient. Nevertheless, these findings unambiguously confirm that excited state electron transfer from [Mn(pbmi)$_2$]$^+$ to the substrate BP is indeed feasible (Fig. 4c). Hence, the photostable

[Mn(pbmi)$_2$]$^+$ complex is thermodynamically and kinetically competent to drive photoreductions paving the way for applications of carbene manganese(I) complexes in photoredox catalysis. The low ground state redox potential, however, renders re-reduction of [Mn(pbmi)$_2$]$^{2+}$ challenging with conventional sacrificial electron donors. Hence, future design strategies aim to shift the ground state redox potential to higher values to enable photoredox catalysis.

We have shown that panchromatic absorption, fast and efficient intersystem crossing, high photostability, and a triplet excited state lifetime of 190 ns are achieved in the manganese(I) complex [Mn(pbmi)$_2$]$^+$ (Fig. 4c) starting from a readily-accessible carbene/pyridine ligand and a simple manganese(II) salt. The enormous increase in $^3$MLCT state lifetime of 3d$^6$ metal complexes over six orders of magnitude from [Fe$^{II}$(bpy)$_3$]$^{2+}$ (~50 fs), over [Fe$^{II}$(pbmi)$_2$]$^{2+}$ derivatives (several ps) and polyisonitrile 3d$^6$ Cr$^0$/Mn$^I$ complexes (~2 ns) to 190 ns in the present [Mn(pbmi)$_2$]$^+$ complex shows that the photophysics and photochemistry of simple coordination complexes incorporating abundant 3d$^6$ metal ions has reached a level on par with complexes of expensive and rare noble metal ions. The present complex represents a big step forward to truly sustainable applications of abundant metals with readily accessible ligands and metal salts including their utilization as sensitizers in synthetic photochemistry, solar fuels production, light-emitting devices, and medicinal photochemistry.

## Methods
### General procedures

[H$_2$pbmi]Br$_2$ and Mn[OTf]$_2$ were obtained from TCI and Sigma-Aldrich, respectively. Other reagents were used as received from commercial suppliers (Sigma-Aldrich, Fisher Sci., TCI, Acros). CH$_3$CN/CD$_3$CN,

diethyl ether, and THF/$d_8$-THF were distilled from $CaH_2$, sodium, and potassium, respectively. Dry acetone, $CH_2Cl_2$ and 2-MeTHF were purchased (AcroSealTM) and degassed by freeze-pump-thaw cycles prior to use. All solvents were stored over pre-activated 3 Å molecular sieve. All reactions and measurements were performed under a water-free argon atmosphere if not stated otherwise. A glovebox (UniLab/Mbraun – Ar 5.0, $O_2 < 0.1$ ppm, $H_2O < 0.1$ ppm) was used to store and weigh sensitive compounds for synthesis as well as to prepare samples for spectroscopic and analytic measurements.

Elemental analysis was performed by the Mikroanalytisches Labor Kolbe, c/o Fraunhofer Institut UMSICHT, Oberhausen, Germany.

NMR spectra were recorded on a Bruker Avance NEO spectrometer at 400 MHz ($^1$H) and 100 MHz ($^{13}C\{^1H\}$). All resonances are reported in ppm versus the solvent signal as an internal standard [$CD_3CN$ ($^1$H: $\delta = 1.94$ ppm), ($^{13}C\{^1H\}$: $\delta = 1.3$ ppm, $\delta = 118.3$ ppm)][51]. Multiplicities are abbreviated as follows: (s) = singlet, (d) = doublet, (t) = triplet.

IR spectra were recorded on an Agilent Cary 630 FTIR spectrometer with an ATR unit containing a diamond crystal inside an argon-filled glovebox. The far-IR spectrum was recorded on a Nicolet 5700 FT-IR spectrometer equipped with a Smart Orbit diamond ATR unit using single crystals covered with Nujol. The intensities are qualitatively indicated with weak (w), medium (m), strong (s) and very strong (vs).

Raman spectra ($\lambda_{exc} = 1064$ nm) were measured on a Nicolet 5700 FT-IR spectrometer combined with a NXR 9650 FT-Raman Module equipped with a 1064 nm laser (laser power 20–1500 mW; resolution 2 cm$^{-1}$), a Microstage microscope, and a NXR Genie Ge-detector using single crystals or crystalline powders in glass capillaries (under inert gas). Raman spectra with excitation wavelengths of $\lambda_{exc} = 785$, 633, and 532 nm were recorded on a Witec Alpha300 spectrometer. The spectra were recorded by laser excitation of single crystals or crystalline powders in glass capillaries under inert atmosphere. The spectrum obtained with excitation at 532 nm was measured in $d_8$-THF solution in a glass capillary under inert atmosphere. The intensities are qualitatively indicated with weak (w), medium (m), and strong (s).

ESI$^+$ mass spectra were recorded on an Agilent 6545 QTOF-MS spectrometer.

UV-vis-NIR absorption spectra were recorded on Agilent Cary 5000 or Jasco V770 spectrometers using 1.00 cm quartz cells (Hellma, Suprasil) equipped with Schott valves.

Variable temperature steady-state emission spectra were recorded with a FLS1000 spectrometer from Edinburgh Instruments equipped with the cooled, NIR sensitive photomultiplier detector N-G09 PMT-1700. A cw-laser RLTMDL-450-1W-3 from Roithner Lasertechnik ($\lambda_{exc} = 450$ nm, $P = 1089$ mW) was employed for excitation at room temperature. Measurements at 77 K were carried out in a liquid nitrogen cooled cryostat, Optistat DN from Oxford Instruments using a xenon arc lamp Xe2 from Edinburgh Instruments. Luminescence decay curves were recorded in the multi-channel scaling mode employing a variable pulsed laser VPL-450 ($\lambda_{exc} = 450$ nm) as excitation source.

Cyclic voltammetry experiments were carried out on a BioLogic SP-200 voltammetric analyzer in an Ar-filled glove box using platinum wires as counter and working electrodes and 0.01 M Ag/AgNO$_3$ as the reference electrode at a scan rate of 100 mV s$^{-1}$ using 0.1 M [$^nBu_4N$][$PF_6$] as supporting electrolyte in $CH_3CN$. Potentials are referenced relative to the ferrocenium/ferrocene couple as internal standard and converted to the SCE reference[37].

UV-vis-NIR spectroelectrochemical experiments were performed using a TSC 1600 Spectro cell from RHD Instruments equipped with a platinum net working electrode (approximate path length 0.43 mm), a glassy carbon counter electrode and a silver wire as pseudo reference electrode and a potentiostat Autolab IMP from Metrohm. A J&M TIDAS S MMS was used as UV-vis-NIR spectrometer, a Hamamatsu L10290 as excitation source.

fs-Transient absorption experiments were conducted using a Helios pump-probe setup from Ultrafast Systems paired with a regeneratively amplified 1030 nm laser (Pharos, Light Conversion, 1030 nm, <175 fs, 2 mJ). The effective laser repetition rate of 1 kHz was set via an internal pulse picker. A small portion of the 1030 nm fundamental was directed to the optical delay line and was subsequently used to generate broadband probe light by focusing the beam onto a sapphire for measurements in the vis-NIR range (450–900 nm). In the UV-vis spectral range (330–500 nm), the second harmonic was focused onto a second sapphire instead of the fundamental. The pump pulse was generated with an optical parametric amplifier (Apollo Y, Ultrafast Systems), and the beam diameter at the sample was adjusted to 100–150 µm at the sample to assure homogeneous excitation of the observation volume, which is defined by the probe diameter (ca. 25 µm). The sample solutions were measured under argon atmosphere in a 1 mm quartz cuvette. To generate spectra that cover the whole spectral region from 350 nm to 900 nm, the UV-vis and vis-NIR part of the transient absorption spectra were recorded separately under identical conditions and were combined by matching the relative band maxima of both spectral regions with the corresponding ns-transient absorption spectra. For the latter, the entire spectral range can be measured simultaneously. Preprocessing of the data, including chirp and baseline correction, has been performed using the Surface Xplorer 4.3.0 software from Ultrafast Systems. The open-source Python based data analysis tool KiMoPack 7.4.9 was employed for global analysis of the transient absorption data[52].

## Investigation of the ultrafast processes, e.g. ISC/IC and vibronic coupling

In order to increase the time resolution, the first few picoseconds upon excitation were measured in a separate experiment with 20 fs step size. The pump diameter was decreased to ca. 50 µm. To correct for coherent artefacts and the chirp, the solvent response has been measured directly after every measurement. The time resolution of the setup in the described configuration amounts to ca. 125–150 fs and was estimated by fitting of the solvent response traces with a Gaussian and its first and second derivatives (Supplementary Fig. 15). To eliminate coherent artefacts from the decay traces, the weighted solvent response was subtracted from the original dataset. For the analysis of the vibronic coupling, the decay traces between 750 nm and 800 nm (5 nm steps) were used as the corresponding oscillations are most pronounced in this spectral region. The decay traces were fitted individually with a triexponential function; the residuals of the resulting fits were averaged and analyzed by FFT. For the determination of the dephasing time $\tau_{dephasing}$, the mean residuals were also fitted with an exponentially decaying cosine function of the form $A\,e^{-t/\tau_{dp}} \cdot \cos(\omega t + \varphi)$. The analysis of the vibronic coherence and the correction of the coherent artefacts was carried out using the software Mathematica 12.0 from Wolfram[53].

ns-Transient absorption spectroscopy experiments were carried out using a modified version of the described fs-transient absorption spectroscopy setup. For this purpose, the Eos add-on has been employed, which uses a photonic crystal fiber based supercontinuum laser as probe light source. In contrast to the fs measurements, the pump-probe time delay is controlled electronically with a time resolution of <1 ns.

Photostability experiments were conducted by irradiating diluted dry, deaerated solutions of the compound in $CH_3CN$ in an inert gas cuvette (optical path length $d = 1$ cm) under constant stirring. A high-power LED from Prizmatix (UHP-T-520-DI; output power: 2.2 W) with an emission maximum at 523 nm was used as excitation light source (Supplementary Fig. 16c). The collimated beam was focused onto the cuvette with a plano convex lens. At the sample position, the beam diameter was adjusted to 0.5 cm. The temperature of the irradiated solution was kept at 293 K with a Peltier module. To monitor the decay

of the complex, the irradiation has been interrupted at several points in time to measure UV-vis absorption spectra (Supplementary Fig. 16a). The corresponding decay trace illustrated in Supplementary Fig. 16 d was obtained using the absorbance at 505 nm.

To estimate the photodegradation quantum yield $\Phi_{deg}$, $[Ru(bpy)_3]^{2+}$, which has a literature-known photodegradation quantum yield ($\Phi_{deg} = 0.022$ %)[32], was employed as actinometer. For this purpose, a diluted solution of $[Ru(bpy)_3][PF_6]_2$ in dry, deaerated $CH_3CN$ has been irradiated as described for the photostability experiments. The complex concentration $c$ was monitored using the phosphorescence band centered at 620 nm (Supplementary Fig. 16b), which was measured with an FS5 emission spectrometer from Edinburgh Instruments. In the case of a monophotonic photodecomposition mechanism, the degradation rate $dc/dt$ of a chromophore is given by the product of the degradation quantum yield and the rate of photon absorption events $I_{abs}$ (Eq. (1)). For diluted solutions ($A < 0.1$), the photon absorption rate depends linearly on the absorbance and can be expressed with Eq. (2), where $\varepsilon$ and $I_0$ denote the molar absorption coefficient and the photon flux, respectively. Combining Eq. (1) and eq. (2), a first order rate law is obtained (Eq. (3)), which in turn yields a monoexponential integrated rate law (Eq. (4)), where $c_0$ and $k_{deg}$ represent the initial concentration of the chromophore and the decay rate constant, respectively.

$$\frac{dc}{dt} = -I_{abs}\,\phi_{deg} \tag{1}$$

$$I_{abs} = I_0\left(1 - 10^{-A}\right) \approx 2.303\,A\,I_0 = 2.303\,\varepsilon c\,d\,I_0 \tag{2}$$

$$\frac{dc}{dt} = -2.303\,\varepsilon\,c\,d\,I_0\,\phi_{deg} \tag{3}$$

$$c = c_0\,e^{-k_{deg}t} = c\,e^{-2.303\,\varepsilon\,d\,I_0\phi_{deg}t} \tag{4}$$

As illustrated by Supplementary Fig. 16d, the complex follows the predicted behavior and decays monoexponentially during irradiation. Thus, for a monochromatic light source, the degradation quantum yield could be estimated with Eq. (5).

$$\phi_{deg,Mn} = \phi_{deg,Ru}\,\frac{k_{deg,Mn}\,\varepsilon_{Ru}}{k_{deg,Ru}\,\varepsilon_{Mn}} \tag{5}$$

However, the UHP-LED emits not only at 523 nm, but also in the vicinity of the central wavelength (Supplementary Fig. 16c). Therefore, Eq. (5) has to be adapted to account for the different absorptivity of the complex and the actinometer weighted with the relative photon flux $I_\lambda$ at all wavelengths $\lambda$ yielding Eq. (6).

$$\phi_{deg,Mn} = \phi_{deg,Ru}\,\frac{k_{deg,Mn}}{k_{deg,Ru}}\,\frac{\sum \varepsilon_{\lambda,Ru}\,I_\lambda}{\sum \varepsilon_{\lambda,Mn}\,I_\lambda} = 0.0002\,\% \tag{6}$$

Quenching experiments with benzophenone were prepared by dissolving 125 mg (0.69 mmol) benzophenone in $CH_3CN$ (3 ml), followed by taking 434.8 μl of this solution and adding it to 481.4 μl of $CH_3CN$ under inert atmosphere. From a 3.34 mM stock solution of $[Mn(pbmi)_2][OTf]$ in $CH_3CN$ 83.8 μl were taken and combined with the benzophenone solution to obtain a solution of $[Mn(pbmi)_2][OTf]$ (280 μM) and benzophenone (100 mM) in $CH_3CN$.

## X-ray crystal structure analysis
Intensity data for crystal structure determination were collected with a STOE STADIVARI diffractometer from STOE & CIE GmbH with an Oxford cooling using Mo-K$_\alpha$ radiation ($\lambda = 0.71073$ Å). The diffraction frames were integrated using the STOE X-Area[54] software package and were corrected for absorption with STOE LANA[55,56] of the STOE X-Area software package by scaling of reflection intensities followed by a spherical absorption correction. The structures were solved with SHELXT[57] and refined by the full-matrix method based on $F^2$ using SHELXL[58] of the SHELX[59] software package and the ShelXle[60] graphical interface. All non-hydrogen atoms were refined anisotropically, while the positions of all hydrogen atoms were generated with appropriate geometric constraints and allowed to ride on their respective parent atoms with fixed isotropic thermal parameters. Crystallographic data for the structure reported in this paper has been deposited with the Cambridge Crystallographic Data Centre as supplementary publication no. CCDC-2396534.

## Synthesis of [Mn(pbmi)$_2$][OTf]
In a flame dried and argon-purged 50 ml Schlenk flask equipped with a stirring bar, $[pbmi]Br_2$ (600 mg, 1.49 mmol, 2.5 eq) was suspended in dry THF (30 ml). The mixture was cooled to 195 K, followed by dropwise addition of sodium bis(trimethylsilyl)amide (1 M in THF, 2.98 ml, 5 eq). After stirring at 195 K for 30 minutes, anhydrous $Mn[OTf]_2$ (210 mg, 0.59 mmol, 1 eq) was added as a solid. The solution was allowed to reach 293 K accompanied by a color change from dark brown to dark purple. After stirring for 16 h in the dark, the formed dark brown precipitate was filtered off and washed with THF (40 ml). The THF was removed under reduced pressure from the filtrate. The resulting crude product was dissolved in acetone (80 ml), the solution was filtered, and the solvent was removed under reduced pressure. The solid was dissolved in $CH_2Cl_2$ (5 ml) and the solution was filtered. Addition of PhCH$_3$ (20 ml) at 255 K formed dark purple needles. Recrystallization from $CH_2Cl_2$/PhCH$_3$ as above was repeated. The purple needles were washed with diethyl ether (3 × 3 ml) and dried under reduced pressure to give $[Mn(pbmi)_2][OTf]$ (160 mg, 0.23 mmol, 39 %).

## Crystallization for single crystal X-ray diffraction
$[Mn(pbmi)_2][OTf]$ (50 mg) was dissolved in THF (5 ml), filtered through a syringe filter, and crystallized by addition of diethyl ether (10 ml) at 255 K giving dark purple crystals suitable for X-ray diffraction.

## Crystallographic data of [Mn(pbmi)$_2$][OTf]
$C_{27}H_{26}F_3MnN_{10}O_3S\times3THF$ (898.89); monoclinic; $P2_1/n$; $a = 15.828(3)$ Å, $b = 15.506(3)$ Å, $c = 17.001(3)$ Å, $\beta = 92.54(3)°$; $V = 4168.4(14)$ Å$^3$; $Z = 4$; density (calculated) 1.432 g cm$^{-3}$, $T = 120(2)$ K, $\mu = 0.439$ mm$^{-1}$, $F(000) = 1880$; crystal size $0.100 \times 0.073 \times 0.040$ mm$^3$; $\theta = 1.778$ to 28.499 deg.; $-21 \leq h \leq 21$, $-17 \leq k \leq 20$, $-22 \leq l \leq 22$; rfln collected = 42399; rfln unique = 10571 [$R$(int) = 0.0972]; completeness to $\theta = 25.242$ deg. = 100.0 %; semi empirical absorption correction from equivalents; max. and min. transmission 0.9798 and 0.6228; data 10571; restraints 995, parameters 756; goodness-of-fit on $F^2 = 1.019$, final indices [$I > 2\sigma(I)$] $R_1 = 0.0873$, $wR_2 = 0.1841$; $R$ indices (all data) $R_1 = 0.2089$, $wR_2 = 0.2479$; largest diff. peak and hole 0.430 and $-0.597$ e Å$^{-3}$.

Elemental analysis: obs. C 47.56, H 3.88, N 20.51; calcd for $C_{27}H_{26}MnN_{10}F_3O_3S$, C 47.51, H 3.84, N 20.52.

$^1$H NMR (400 MHz, CD$_3$CN): $\delta$ / ppm = 7.98 (s, 4 H, H$^2$), 7.59 (d, $^3J_{HH} = 7.8$ Hz, 4 H, H$^6$), 7.47 (t, $^3J_{HH} = 7.8$ Hz, 2 H, H$^7$), 6.77 (s, 4 H, H$^3$), 2.16 (s, 12 H, H$^1$).

$^{13}$C{$^1$H} NMR (100 MHz, CD$_3$CN): $\delta$ / ppm = 218.6 (s, C$^4$), 150.5 (s, C$^5$), 125.1 (s, C$^3$), 123.3 (s, C$^7$), 114.5 (s, C$^2$), 101.3 (s, C$^6$), 34.7 (s, C$^1$).

ATR-IR: $\tilde{\nu}$ / cm$^{-1}$ = 1617 (w), 1485 (m), 1468 (m), 1438 (m, sh), 1395 (m), 1325 (m), 1276 (m, sh), 1256 (s, triflate), 1226 (m), 1153 (m, triflate), 1073 (m), 1028 (s, triflate), 997 (m), 939 (w), 775 (w), 756 (m, triflate), 715 (m), 672 (s), 650 (m), 635 (vs, triflate), 579 (m, triflate), 516 (m), 465 (m), 400 (m), 326 (m), 317 (m).

Raman ($\lambda_{exc} = 1064$ nm, solid): $\tilde{\nu}$ / cm$^{-1}$ = 3149 (w), 3088 (w), 2944 (w), 1596 (s), 1554 (w), 1483 (w), 1428 (w), 1402 (w), 1342 (w), 1321 (w,

sh), 1236 (w, sh), 1230 (w, sh), 1224 (w), 1170 (w), 1134 (w, sh), 1124 (w), 1089 (s, sh), 1084 (s), 1068 (s), 1064 (s), 1028 (w), 1019 (w), 1002 (w), 755 (w, triflate), 710 (m), 655 (w), 514 (s), 378 (s), 124 (m).

ESI$^+$ (CH$_3$CN): $m/z$ (%) = 682.12 (2.3, {[Mn(pbmi)$_2$][OTf]}$^+$), 533.17 (100, [Mn(pbmi)$_2$]$^+$), 266.59 (2.2, [Mn(pbmi)$_2$]$^{2+}$).

HR-ESI$^+$ (CH$_3$CN): $m/z$ (%) = 533.1724 (100). Calcd. for [C$_{26}$H$_{26}$MnN$_{10}$]$^+$: $m/z$ (%) = 533.1717.

UV-vis-NIR (CH$_3$CN): $\lambda_{max}$ / nm ($\varepsilon$ / $10^3$ M$^{-1}$ cm$^{-1}$) = 575 (16.5), 505 (23.0), 362 (16.7), 286 (24.5), 243 (40.0).

UV-vis-NIR (THF): $\lambda_{max}$ / nm = 363, 509, 579.

UV-vis-NIR (CH$_2$Cl$_2$): $\lambda_{max}$ / nm = 364, 511, 580.

Emission (CH$_3$CN, $\lambda_{exc}$ = 450 nm, 293 K): $\lambda_{em}$ = 1025 nm; $\tau_{PL}$ = 190 ns.

Emission (2-MeTHF, $\lambda_{exc}$ = 505 nm, 77 K): $\lambda_{em}$ = 980, 1012, 1090 nm; $\tau_{PL}$ = 7.1 μs.

CV (CH$_3$CN/[$^n$Bu$_4$N][PF$_6$]): $E_{1/2}$ / V vs ferrocenium/ferrocene = −0.96, −0.22, 0.99. ($E_{1/2}$ / V vs SCE = −0.58, 0.16, 1.37).

Density Functional Theory (DFT) calculations were carried out using the ORCA program package[61] (versions 5.0.3 or 5.04). Tight convergence criteria were chosen for all calculations (keywords *tightscf* and *tightopt*). All calculations were performed using the B3LYP functional[62–64] using Ahlrichs' polarized valence triple-zeta basis set (def2-TZVP)[65,66] employing the RIJCOSX approximation (keyword *RIJCOSX*)[67,68]. Relativistic effects were calculated at the zeroth order regular approximation (keyword *ZORA*) level[65] using relativistically adjusted basis sets. To account for solvent effects, a conductor-like screening model (keyword *CPCM*) modeling acetonitrile was used in all calculations[69,70]. Geometry optimizations were performed Atom-pairwise dispersion correction was performed with the Becke-Johnson damping scheme (keyword *D3BJ*)[71,72]. Explicit counter ions and/or solvent molecules were not taken into account. The reported calculated IR and Raman frequencies were obtained from a numerical frequency calculation without a solvent model, and they were scaled by a factor of 0.98. The charge transfer number analyses of the fifty time-dependent DFT (TDDFT)-calculated transitions (keyword *nroots 50*) were done using TheoDORE 2.2[73,74]. The reported calculated transition energies are shifted by 0.33 eV to lower energies (Supplementary Table 1). For charge transfer number analysis, the complex cation was divided into three fragments: the manganese centre, the two pyridines, and the four carbenes. For SOC-TDDFT calculations of the radiative rate constants $k_r$ of the triplet sublevels, the keywords *triplets true*, *dosoc true*, *tda false,* and RI-SOMF(1X) were used, and the data was evaluated according to refs. 41,42.

## Data availability

Source Data are provided with this manuscript. All pertinent experimental procedures, materials, methods, and characterization data (NMR spectroscopy, electrospray ionization mass spectrometry, X-ray diffraction, optical spectroscopic, and electrochemical data as well as xyz coordinates of calculated geometries) are provided in this article, the Supplementary Information, and under https://doi.org/10.6084/m9.figshare.29155433. The X-ray crystallographic coordinates for structures reported in this study have been deposited at the Cambridge Crystallographic Data Centre (CCDC), under deposition number 2396534. These data can be obtained free of charge from The Cambridge Crystallographic Data Centre via www.ccdc.cam.ac.uk/data_request/cif. All data are available from the corresponding author upon request. Source data are provided with this paper.

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

## Acknowledgements

This work was supported by the Max Planck Graduate Center with the Johannes Gutenberg University Mainz (MPGC) by a stipend to S. K. This work was further supported by the Deutsche Forschungsgemeinschaft through grants INST 247/1018-1 FUGG and INST 247/1082-1 FUGG to K. H. We thank Dr. Dieter Schollmeyer (Johannes Gutenberg University Mainz, Germany) for collecting the XRD data and Dr. Detlev-Walter Scholdei (Max Planck Institute for Polymer Research, Mainz, Germany) for recording Raman spectra with visible light excitation. Parts of this research were conducted using the supercomputer Elwetritsch and advisory services offered by the Rheinland-Pfälzische Technische Universität Kaiserslautern-Landau (https://hpc.rz.rptu.de), which is a member of the Allianz für Hochleistungsrechnen Rheinland-Pfalz (AHRP). The funders had no role in study design, data collection and analysis, decision to publish, or preparation of the paper.

## Author contributions

S.K. carried out the synthetic, spectroscopic, electrochemical and computational work and analyzed the data, R.N. provided guidance in the spectroscopic work, designed the photochemical studies and helped in data analysis, C.F. performed and provided guidance in the quantum chemical analysis, performed the calculation of the radiative rate constants and solved the XRD structure, N.R.E. performed the initial synthesis and preliminary characterization of the complex, J.K. performed the far-IR, Raman and resonance Raman studies, and K.H. conceived the project, wrote the manuscript and provided guidance. All the authors contributed to the writing and editing of the manuscript and gave approval to its final version.

## Funding

## Competing interests

The authors declare no competing interests.
