## [Transparent Peer Review file · Nature Communications]

A manganese(I) complex with a 190 ns metal-to-ligand charge transfer lifetime

Corresponding Author: Professor Katja Heinze

Version 0:

Reviewer comments:

Reviewer #1

(Remarks to the Author)

The authors present a tetracarbene manganese(I) complex which phosphoresces at room temperature. The manuscript provides good characterization of the Mn(II/I) redox couple and the ability of the Mn excited state to reduce an organic substrate via bimolecular quenching. While the excited state lifetime is very long lived for a 3d⁶ complex, so it is presumably a high energy excited state, the manuscript characterizes this as an MLCT.

The largest issue seems to be that all of the calculations show that all the excitations are very mixed MLCT, LC, and MC in nature (Table S1 figure). Can this excited state really be assigned as an MLCT? More evidence and discussion of this is needed in the main manuscript before publication.

In addition, Table S1 shows TDDFT excited states but Figure 2 doesn't show any of the actual transitions that the difference orbitals correspond to. The differences are not for the experimental absorption peak so the current figure is misleading.

In Figure 2 the difference orbitals are very small, and it is hard to see where the density is on the ligands. In addition, all the iso values show very little density for all the orbitals (including the base MOs in Table S7). This should be commented on or changed.

The computational methods are very unclear.

1. In the methods section says "The excited singlet states 1MLCT(n) (keyword iroot n) with MLCT character were geometry optimized using the keyword nroots 10..." but in the Figure 2 caption the absorption orbitals are called these names.

2. The method functional and basis set listed in Figure 2 caption is missing the dispersion correction and solvation model discussed in the methods. Please list the computational method in each caption that includes computational results.

If indeed the 1MLCT(n) states are optimized geometries or orbitals these should be characterized in the SI not just the vertical excitations in Table S1.

The optimized structures should be characterized structurally. In the discussion of not being able to find a 3MC optimized structure the authors should indicate the structural space explored (how much were the bonds expanded?) and if those structures while not optimized were MC in character on the triplet surface.

What would constitute an MC in this kind of carbene system? The initial transitions in Table S1 all have some ligand character. This should be discussed in the text as it is critical for these classes of complexes.

Was a crossing point between the S₀ and T₁ surface found?

For the fitting of the time-resolved spectroscopy. The longest global fit lifetime is 190 ns but the full excited state doesn't seem to decay until 1500 ns, is the data better fit by and additional lifetime?

Other things I noticed:

Make sure acronyms and naming conventions are defined at the first instance not later in the methods without a reference. "fluid solution" sounds strange to me. I think liquid or just in solution would be simpler

Reviewer #2

(Remarks to the Author)

The manuscript by Kronenberger describes the synthesis and photophysical investigation of a new d⁶ manganese(I) complex which exhibits near-IR phosphorescence from a triplet MLCT state with a lifetime of 190 ns which begins to rival the lifetimes of more established but less sustainable precious metal complexes based on d⁶ ruthenium(II) ions. Whilst the photoluminescent quantum yield is quite low due to a slow rate of radiative deactivation, the work presents a major step forward in the development of sustainable photosensitisers based on Earth abundant metal elements. The work appears to have been meticulously carried out and is reported in a clear and coherent manner and could therefore be accepted for

publication subject to minor amendments.

The authors argue that the long lifetime and photostability stem from high energy 3MC states not being accessible and relevant to non-radiative deactivation of the complex due to a high degree of covalency in Mn-L sigma bonding compared to the Fe(II) analogue where the 3MC state plays a dominant role in its photophysical behaviour. Whilst this might be in part true, is it not also due to the reduced oxidation state and lower nuclear and overall charge which results in higher energy d orbitals for Mn(I) vs Fe(II) relative to ligand orbitals? This is the principal reason that the 1MLCT absorption bands are at much lower energy than for Fe(II). But this also means that the set of both e_g^* and t_{2g} orbitals are shifted upwards relative to the ligand orbitals, thus this will destabilise 3MC vs the 3MLCT. This is essentially what the manuscript is saying but from an alternative vantage point but the manuscript perhaps doesn't make enough of the what is a very interesting design strategy in exploiting the higher energy d-orbitals of a lower oxidation state isoelectronic ion to achieve what the analogous iron(II) complex cannot. The authors might consider slightly augmenting the discussion here.

From the ESI, the lowest energy 1MC states are determined. The manuscript states that 3MC states can't be located on the T1 surface by U-DFT optimisation from appropriate guess geometries, but have the energies of 3MC states been determined from TDDFT calculations? These would be worth a brief mention in the manuscript.

Electrochemical voltages relative to ferrocene should be added, at least in parentheses for ease of comparison to other literature where this reference couple is ubiquitous.

Page 4, line 32 – the Franck-Condon excited state dipole moments being similar to that of the ground state... shouldn't the dipole of the ground state be zero given the symmetry of the complex?

The colour coding in Fig S11 is a little difficult to make out. Could the colours be adapted to make differentiation easier?

In summary, a very nice piece of work.

Reviewer #3

(Remarks to the Author)

The manuscript by Kronenberger et. al. describes the synthesis and characterization of a novel Mn(I) complex with mixed NHC/pyridine ligands. The main finding of this study is the remarkably long (190 ns) lifetime of a weakly emissive excited state that is identified as 3MLCT state and was shown to engage in bimolecular electron transfer. The very photostable tetracarbene complex is based on the well-known tridentate pbmi ligand and is readily obtained from a simple Mn(II) salt and the commercially available pro-ligand. The gain in excited state lifetime is a major leap forward compared to the isoelectronic Fe(II) complexes based on this ligand or variations thereof (tens of ps) or other d6 base metal (Fe(II), Mn(I), Cr(0)) complexes with excited state lifetimes on the order of a few nanoseconds.

Identification of the long-lived excited state as 3MLCT is convincing despite the very low emission quantum yield that translates into a radiative rate constant more expected for a transition that is both spin and symmetry forbidden, like for a 3MC state. The slow radiative decay of the 3MLCT is however rationalized by quantum chemical calculations that reveal a rather weak spin orbit coupling. Furthermore, these calculations indicate that the slow non-radiative decay of the 3MLCT can be attributed to the absence of any lower-lying MC states.

With the results described in this very well-written manuscript the authors convincingly demonstrate that complexes of abundant metals truly hold great potential as sustainable alternatives to the dominating precious metal photosensitizers in important applications such as synthetic photochemistry or chemical solar energy conversion. I have therefore no hesitation in recommending this manuscript for publication in Nature Communications.

The authors should however consider the following points:

- 1) The remarkable lifetime and photostability of the Mn complex are attributed to the exceptional destabilization of the 3MC state making it inaccessible from the 3MLCT state. It is argued that N-Mn-N elongation that would stabilize a 3MC state is restricted. Do the authors have experimental or computational support for this restriction? Could the potential energy surface be calculated?
- 2) The voltammetric data in Figure 1 does not extend to potentials below -1.6 V vs SCE and no ligand reduction can be observed in this range. The available potential window in acetonitrile does however extend much more negative, and differential pulse voltammograms can be recorded to at least -2.6 V vs SCE. It would be rewarding to compare the potential for ligand reduction to the expected value around -2.5 V based on the potential of the Mn(II/I) couple and the MLCT excitation energy.
- 3) The assignment of the quasi-reversible wave with $E_{1/2} = 1.37$ V to the Mn(IV/III) redox couple is questionable. The lowest energy absorption band of the Mn(II) state, presumably due to LMCT excitation, is suggesting that oxidation of the ligand occurs at about 1.4 V.
- 4) The oxidative quenching with benzoquinone seems to suffer from a relatively low cage escape yield. Assuming a typical excited state concentration on the order of 10^{-5} M and considering the efficient (> 90 % quenching), the observed transient absorption corresponds to a cage escape yield on the order of a few percent (on the order of 10^{-7} M BQ $^-$, with $\epsilon \sim 10^4$ M $^{-1}$ cm $^{-1}$). Could the authors provide more precise information regarding the cage escape yield and comment on what seems to be a rather small value?
- 5) With a potential of -0.58 V vs. SCE for the Mn(II/I) couple, I could imagine that the Mn(I) complex is quite easily oxidized by atmospheric oxygen. Could the authors comment on this potential complication?
- 6) For any catalytic application exploiting the relatively strongly reducing excited state of the Mn(I) complex, the photoactive state would have to be regenerated with a suitable electron donor. The very weakly oxidizing Mn(II) state (-0.58 V vs. SCE) would however severely limit the choice of suitable electron donors. Could the authors comment on how this would affect potential applications?

Version 1:

Reviewer comments:

Reviewer #1

(Remarks to the Author)

The authors have substantially revised the manuscript and answered all questions well in their response letter. More discussion about the excited states' character in the manuscript would strengthen the paper. These would include the quantification of the LC vs CT in the initial excitations, the discussion of the vertical MC excitations and their energies. With these and the added text of the optimized MC the authors could make a broader statement of how Mn is an ideal metal for MC destabilization and longer-lived excited states.

Reviewer #2

(Remarks to the Author)

Many thanks to the authors for their positive engagement with suggested changes and queries. I am satisfied with the changes made and can see that the authors have similarly made diligent efforts to meet the expectations of the other referees. I therefore recommend publication of the current manuscript without change.

Reviewer #3

(Remarks to the Author)

The authors have satisfactorily addressed most of my concerns and I recommend publication in Nature Communications. I however believe that the following points should be considered:

- 1) The additional electrochemical data (SI, Fig. 6) should be complemented with the voltammogram of the electrolyte background if the authors want to attribute the onset of cathodic current to the reduction of the complex.
- 2) The practical limitations (with regard to applications in e.g. photoredox catalysis) emerging from the exceptionally low ground state potential of the Mn(II/I) couple (points 5 and 6 of my original review) would deserve more attention (e.g. at the end of the "Bimolecular photoinduced electron transfer" paragraph) to give the reader a better perspective on the remaining challenges.

We have addressed the points raised the reviewers as follows:

Reviewer 1:

The largest issue seems to be that all of the calculations show that all the excitations are very mixed MLCT, LC, and MC in nature (Table S1 figure). Can this excited state really be assigned as an MLCT?

This reviewer is correct that the excited states are quite mixed in nature according to the calculation. This is quite common in transition metal complexes, in particular when metal-ligand bonding becomes more covalent as is the case here. However, the characteristic transitions #5 and #11, for example, possess more than 50% MLCT character (see Supplementary Table S1). The next large contribution arises from LLCT (<30%). For this reason, we chose to label the transitions as MLCT, while being fully aware that all transitions are mixed. We added this information to the text as follows: “Ligand-to-ligand charge transfer (LLCT) character also contributes to these excited states, but to a smaller extend (Supplementary Information Table 1).”

In addition, Table S1 shows TDDFT excited states but Figure 2 doesn't show any of the actual transitions that the difference orbitals correspond to. The differences are not for the experimental absorption peak so the current figure is misleading.

This reviewer is correct that we missed to add the calculated transitions/oscillator strengths to Fig. 2a. This has now been corrected and we added the following text to the figure caption “TDDFT-calculated oscillator strengths (black, shifted by 0.33 eV to lower energies; B3LYP/Def2-TZVP/CPCM(acetonitrile)/D3BJ)”. We thank the reviewer for spotting this omission.

In Figure 2 the difference orbitals are very small, and it is hard to see where the density is on the ligands. In addition, all the iso values show very little density for all the orbitals (including the base MOs in Table S7). This should be commented on or changed.

In Fig. 2a, we increased the overall size of difference density and spin density plots, but maintained the isosurface values (0.003 is a value typically used). A smaller isosurface value does not change the pictorial description and the statement, that the contributions of Mn and the pyridines are most relevant, but leads to crowding and makes the plot rather illegible.

The computational methods are very unclear. 1. In the methods section says “The excited singlet states 1MLCT(n) (keyword iroot n) with MLCT character were geometry optimized using the keyword nroots 10...” but in the Figure 2 caption the absorption orbitals are called these names.

We thank the reviewer for spotting this mistake in the “Density Functional Theory (DFT) calculations” section. We removed this part.

2. The method functional and basis set listed in Figure 2 caption is missing the dispersion correction and solvation model discussed in the methods. Please list the computational method in each caption that includes computational results.

We added this information to Fig. 2 “B3LYP/Def2-TZVP/CPCM(acetonitrile)/D3BJ” as suggested.

If indeed the $^1\text{MLCT}(n)$ states are optimized geometries or orbitals these should be characterized in the SI not just the vertical excitations in Table S1. The optimized structures should be characterized structurally.

We did not optimize the $^1\text{MLCT}(n)$ states (see above).

In the discussion of not being able to find a ^3MC optimized structure the authors should indicate the structural space explored (how much were the bonds expanded?) and if those structures while not optimized were MC in character on the triplet surface. Was a crossing point between the S_0 and T_1 surface found?

Our attempts to find ^3MC states computationally were as follows:

- When starting from the optimized ^3MC geometry of the isoelectronic iron(II) complex with elongated metal-N bonds (ref. 48), the optimization converged to the $^3\text{MLCT}$ state (optimization with constraints).
- Elongation of two Mn-C bond lengths up to 3 Å or of the Mn-N bonds up to 2.5 Å (optimization with constraints) leads to a new ^3CT state that can be described as manganese(II) intermediate spin ($t_{2g}^4e_g^1$) antiferromagnetically coupled to a ligand radical. Hence, these triplet states are rather charge transfer states (^3CT) than metal centered states (^3MC). After optimization (leading to one elongated Mn-N bond of 2.376 Å), this state is higher in energy than the lowest energy $^3\text{MLCT}$ state **by 0.59 eV**. Furthermore, this state corresponds to a doubly excited configuration, which would have only small SOC matrix elements with the singlet ground state (see e.g. *Annu. Rev. Phys. Chem.* **2021**, 72, 617). Hence, we assume that this ^3CT state plays no role in the excited state decay.

What would constitute an MC in this kind of carbene system? The initial transitions in Table S1 all have some ligand character. This should be discussed in the text as it is critical for these classes of complexes. Inspection of the character of the lowest energy spin-allowed transitions in Supplementary Information Table 1 shows that Franck-Condon states with large ^1MC character (45.9%) are states #12 and #13. It is conceivable that the corresponding ^3MC states possess a similar orbital occupation, i.e. occupation of a d_{z^2} -like metal centered orbital (see blue density plots).

Plot of difference density

plot of electron gain only

(from Supplementary Information Table 1)

We added these information on the triplet states to the main text as follows:

Instead, a distorted triplet charge-transfer state (^3CT) with a Mn–N bond length of 2.376 Å described by a manganese(II) intermediate spin ($t_{2g}^4e_g^1$) antiferromagnetically coupled to a ligand radical could be localized 0.59 eV above the lowest energy $^3\text{MLCT}$ state. In addition to the high energy, the doubly

excited configuration of this ^3CT state prevents strong coupling with the ground state. Hence, this state will not contribute to the excited state decay.

We thank the reviewer for putting our attention to this point.

For the fitting of the time-resolved spectroscopy. The longest global fit lifetime is 190 ns but the full excited state doesn't seem to decay until 1500 ns, is the data better fit by an additional lifetime?

No, this is not the case. Both the decay kinetics observed by transition absorption spectroscopy and observed by PL spectroscopy can be fitted by a single exponential on the nanosecond time scale. There is no residual at longer times, just the remainder exponential tail, hence the ground state recovery is complete. Please see for example the inset in Fig. 3b, that confirms the monoexponential fit:

Other things I noticed:

Make sure acronyms and naming conventions are defined at the first instance not later in the methods without a reference.

We now defined XRD and IR in the caption of Figure 1 (first occurrence).

“fluid solution” sounds strange to me. I think liquid or just in solution would be simpler

We use the term “fluid solution” to distinguish from “frozen solution”, which has profound implications for excited state decay. As we stated the temperature (293 K), “fluid” is indeed redundant and we removed “fluid”.

Reviewer 2:

We thank the reviewer for the very positive assessment.

Whilst this might be in part true, is it not also due to the reduced oxidation state and lower nuclear and overall charge which results in higher energy d orbitals for Mn(I) vs Fe(II) relative to ligand orbitals? This is the principal reason that the 1MLCT absorption bands are at much lower energy than for Fe(II). But this also means that the set of both e_g^* and t_{2g} orbitals are shifted upwards relative to the ligand orbitals, thus this will destabilise 3MC vs the 3MLCT. This is essentially what the manuscript is saying but from an alternative vantage point but the manuscript perhaps doesn't make enough of the what is a very interesting design strategy in exploiting the higher energy d-orbitals of a lower oxidation state isoelectronic ion to achieve what the analogous iron(II) complex cannot. The authors might consider slightly augmenting the discussion here.

This reviewer is perfectly right, that the lower oxidation state of Mn(I) compared to Fe(II) destabilizes all d orbitals reducing the ³MLCT energy and hence increases the ³MC / ³MLCT energy gap.

We added this explanation to the text as follows on page 5: Compared with the iron(II) complex,²² the higher energy d orbitals of manganese(I) explain its lower energy MLCT states. Compared with the polyisocyanide manganese(I) complexes,²⁰ the lower energy π^* orbital of the pyridine explains the lower energy MLCT states.

And we summarized on page 7:

Together with the low energy ³MLCT states thanks to the high energy d orbitals of manganese(I), the ³MLCT/³MC energy gap is large and excited state decay via ³MC states is of little importance.

We thank the reviewer for bringing our attention to this point.

From the ESI, the lowest energy 1MC states are determined. The manuscript states that 3MC states can't be located on the T1 surface by U-DFT optimisation from appropriate guess geometries, but have the energies of 3MC states been determined from TDDFT calculations? These would be worth a brief mention in the manuscript.

The search for ³MC states has been described above for reviewer 1. And this has been mentioned in the manuscript as suggested.

Electrochemical voltages relative to ferrocene should be added, at least in parentheses for ease of comparison to other literature where this reference couple is ubiquitous.

These values (referenced against ferrocene) were already available in the Methods section:

CV (CH₃CN/[ⁿBu₄N][PF₆]): $E_{1/2}$ / V vs ferrocenium/ferrocene = -0.96, -0.22, 0.99. ($E_{1/2}$ / V vs SCE = -0.58, 0.16, 0.37).

We would like to keep a single reference in the manuscript text to avoid confusion with too many numbers.

Page 4, line 32 – the Franck-Condon excited state dipole moments being similar to that of the ground state... shouldn't the dipole of the ground state be zero given the symmetry of the complex?

This is perfectly true. We have deleted this statement from the text.

The colour coding in Fig S11 is a little difficult to make out. Could the colours be adapted to make differentiation easier?

This is perfectly true. We have adapted the colors as follows:

Reviewer 3:

We thank the reviewer for the very positive evaluation.

1) The remarkable lifetime and photostability of the Mn complex are attributed to the exceptional destabilization of the ³MC state making it inaccessible from the ³MLCT state. It is argued that N-Mn-N elongation that would stabilize a ³MC state is restricted. Do the authors have experimental or computational support for this restriction? Could the potential energy surface be calculated? As suggested by reviewer 1, we had made restricted DFT geometry optimizations with elongated Mn-C and Mn-N bonds. These calculations lead merely to a new ³CT state (with a doubly excited configuration) but not to true ³MC states. This in combination with the low energy ³MLCT states leading to large ³MLCT/³MC gaps explains the long lifetime.

2) The voltammetric data in Figure 1 does not extend to potentials below -1.6 V vs SCE and no ligand reduction can be observed in this range. The available potential window in acetonitrile does however extend much more negative, and differential pulse voltammograms can be recorded to at least -2.6 V vs SCE. It would be rewarding to compare the potential for ligand reduction to the expected value around -2.5 V based on the potential of the Mn(II/I) couple and the MLCT excitation energy. We thank this reviewer for the suggestion. We measured the cyclic voltammogram up to -2.6 V and found the onset of the ligand reduction. This was added to the main text:"

" , while a reductive event was detected close to the solvent potential window (ca. -2.6 V, Supplementary Fig. 6)" and the Supplementary Information (new Fig. 6):

3) The assignment of the quasi-reversible wave with $E_{1/2} = 1.37$ V to the Mn(IV/III) redox couple is questionable. The lowest energy absorption band of the Mn(II) state, presumably due to LMCT excitation, is suggesting that oxidation of the ligand occurs at about 1.4 V. This is an interesting question. Indeed, the lowest absorption of the Mn(II) complex is due to MLCT, not LMCT transitions according to TDDFT calculations. Similarly, the spin density of the optimized tetracation $[\text{Mn}(\text{pbmi})_2]^{4+}$ is fully localized at the manganese center, suggesting a metal centered oxidation. This behavior is analogous to other manganese complexes with strongly donating ligands that stabilize high oxidation states, see e.g. *Nat. Chem.* **2024**, *16*, 827 and *Inorg. Chem.* **2022**, *61*, 14616. As we will report the properties of the other charge states of $[\text{Mn}(\text{pbmi})_2]^{n+}$ in a future contribution, this will not be discussed in the present paper.

4) The oxidative quenching with benzoquinone seems to suffer from a relatively low cage escape yield. Assuming a typical excited state concentration on the order of 10^{-5} M and considering the efficient (> 90 % quenching), the observed transient absorption corresponds to a cage escape yield on the order of a few percent (on the order of 10^{-7} M BQ⁻, with $\epsilon \sim 10^4$ M⁻¹ cm⁻¹). Could the authors provide more precise information regarding the cage escape yield and comment on what seems to be a rather small value?

The reviewer is perfectly right. The cage escape quantum yield with benzophenone is very small. In fact, we tried to quantify the cage escape by laser flash photolysis experiments but failed to provide reasonable values (probably less than 3%). In fact, individual impact of the many factors that could

govern the cage escape field cannot be determined from a single experiment (see *Chem. Rev.* **2024**, *124*, 7379). Hence, we would refrain from a detailed discussion and merely state the observation.

"Yet, the intensity of the BP*⁻ absorption suggests that cage escape is very inefficient."

5) With a potential of -0.58 V vs. SCE for the Mn(II/I) couple, I could imagine that the Mn(I) complex is quite easily oxidized by atmospheric oxygen. Could the authors comment on this potential complication?

The reviewer is correct, the Mn(I) complex is easily oxidized under air and requires anaerobic conditions for handling. This is stated in all experimental procedures (synthesis, quenching).

6) For any catalytic application exploiting the relatively strongly reducing excited state of the Mn(I) complex, the photoactive state would have to be regenerated with a suitable electron donor. The very weakly oxidizing Mn(II) state (-0.58 V vs. SCE) would however severely limit the choice of suitable electron donors. Could the authors comment on how this would affect potential applications?

The reviewer is correct. Suitable electron donors are very limited for the present complex. For catalytic reactions requiring rapid regeneration of the oxidized photocatalyst, these potentials would need to be tuned by modification of the complex. In fact, this is subject to current studies.

Reviewer 1

We have added the details of the composition of MLCT transitions #5 and #11 as well as the MC transitions #12 and #13 (pages 4/7):

“Ligand-to-ligand charge transfer (¹LLCT) character also contributes to these excited states, but to a smaller extent (¹MLCT(11) and ¹MLCT(5) with 53.6%/27.6% and 53.9%/29.8% MLCT/LLCT character, respectively; Supplementary Information Table 1).”

We have included the details on the ³MC search (bond elongations), as suggested (page 7):

“Indeed, ³MC states (as described by a single electron transfer from a d_{xz} , d_{yz} or d_{xy} orbital to a d_{z^2} or $d_{x^2-y^2}$ orbital; see Supplementary Information Table 1 for corresponding ¹MC states, transitions 12 and 13 both with 45.9% MC character) could not be localized by DFT calculations even with induced distortions along the manganese-ligand bonds (Mn–N up to 2.5 Å; Mn–C up to 3.0 Å).”

Reviewer 2

N/A

Reviewer 3

- 1) We have added the electrolyte background (Fig. S6b, Supplementary Information). This suggests that the reduction process is outside the available scan range.
- 2) We have added a comment to illustrate the challenge (page 9): “The low ground state redox potential, however, renders re-reduction of $[\text{Mn}(\text{pbmi})_2]^{2+}$ challenging with conventional sacrificial electron donors. Hence, future design strategies aim to shift the ground state redox potential to higher values to enable photoredox catalysis.”